# Association between proton pump inhibitor use and risk of fracture: A population-based case-control study

Jong Joo Kim[1], Eun Jin Jang[2], Jiwon Park[3], Hyun Soon Sohn[4]*

1 Pharmaceutical Information Research Institute, Cha University, Gyeonggi-do, Republic of Korea, 2 College of Natural Science, Andong National University, Gyeongsangbuk-do, Republic of Korea, 3 College of Natural Science, Kyungpook National University, Daegu-si, Republic of Korea, 4 College of Pharmacy, Cha University, Gyeonggi-do, Republic of Korea

☉ These authors contributed equally to this work.
* sohn64@cha.ac.kr

**Data Availability Statement:** The data underlying the results presented in the study are available from the National Health Insurance Sharing Service at https://nhiss.nhis.or.kr/bd/ab/bdabb006iv.do.

## Abstract

### Objectives

The purpose of this study was to reconfirm the association between the risk of fracture and proton pump inhibitor use and to establish evidence for defining a high-risk group of patients among proton pump inhibitor users.

### Methods

A nested case-control study was performed using data from the National Health Insurance Sharing Service database from the period January 2007 to December 2017. The study population included elderly women aged ≥65 years with claims for peptic ulcer or gastro-esophageal reflux disease. The cases were all incidental osteoporotic fractures, and up to two controls were matched to each case by age, osteoporosis, and Charlson comorbidity index. Conditional logistic regression was used to calculate the adjusted odds ratio and 95% confidence interval (CI).

### Results

A total of 21,754 cases were identified, and 43,508 controls were matched to the cases. The adjusted odds ratio of osteoporotic fractures related to the use of proton pump inhibitors was 1.15 (95% CI: 1.11–1.20). There was a statistically significant interaction between proton pump inhibitor and bisphosphonate use (p<0.01). The risk of fracture in patients using proton pump inhibitors was 1.15 (95% CI: 1.08–1.92) in bisphosphonate users and 1.11 (95% CI: 1.03–1.20) in bisphosphonate non-users.

### Conclusion

Concomitant use of bisphosphonates and proton pump inhibitors will likely increase the risk of osteoporotic fractures in women aged 65 and over, and caution should be exercised in this high-risk group of patients.

Upon an individual researcher's data set request, NHIS provides customized data to the researcher.

**Funding:** HS Sohn received a grant of the National Research Foundation of Korea (Project No.: 2016R1D1A1B03934390; URL: https://www.nrf.re. kr/eng/index). The funder had no role in study design, data collection and analysis, decision to publish, or preparation of the manuscript.

**Competing interests:** The authors have declared that no competing interests exist.

## Introduction

Proton pump inhibitors (PPIs) are effective gastric-acid-suppressing medications used for the treatment of various gastrointestinal disorders, such as gastrointestinal ulcers, esophagitis, hyperacidity, *Helicobacter pylori* infection, and gastro esophageal reflux disease (GERD) [1, 2]. In Korea, the number of patients using PPIs has increased annually with the 2013 rate being 11.5 times that of the 2003 rate [3].

The increase in PPI usage may be associated with an increase in the number of indicated patients. According to data from the National Health Insurance of Korea, the number of patients treated for peptic ulcer or GERD, the main indications for PPIs, has increased by about 13% over the last 8 years (2017 vs. 2010) [4]. Whilst the incidence of peptic ulcer has decreased, GERD, which has increased 1.5-fold over the period from 2010 to 2017, appears to have contributed significantly to this trend [4]. GERD is a chronic disease that is prone to recurrence, and symptoms can improve or worsen; thus, some patients require long-term treatments. The increase in patients with GERD has resulted in an increase in the number of patients using PPIs, as well as the duration of their administration [3]. In addition, the use of PPIs is recommended to prevent gastrointestinal bleeding complications in patients who use two or more antiplatelet drugs [5], and this has contributed to the continuous increase in the long-term use of PPIs, particularly among elderly patients.

Bisphosphonate (BP), commonly used to prevent osteoporotic fractures, has a major adverse effect on the upper gastrointestinal tract, including esophageal inflammation, ulceration, and dyspepsia [6–10]. Hence, it is expected that BP is administered in combination with a PPI to prevent or treat the adverse events [11]. Identifying whether the association between PPI use and the risk of fractures depends on BP use may provide clinical evidence that informs drug selection and improves the anti-fracture effect of BP and the safety of PPI use.

Epidemiological studies have been reported on the association between risk of fractures and PPI use [10, 12–14]. In particular, some studies have reported higher risk of fracture in BP users due to the interaction of BP and PPIs [15, 16]. These associations differ among races, indicating a higher risk of fracture in Asians than in Europeans (pooled odds ratio (OR): 1.75 vs. 1.42) [15]. One study that demonstrated the interaction between BP and PPIs in Asians was a case-control study conducted by Lee et al. in Korea [16]. In this study, the aOR was 1.30 (95% confidence interval (CI): 1.19–1.42) in BP non-users, which was significantly different from the 1.71 (95% CI: 1.31–2.23) found in BP users, and only BP users showed a trend of increasing risk with a cumulative PPI dose [16]. However, a study done by Itoh et al. performed in Japan showed different results [6]. It showed that BP administration in combination with PPIs may be more effective not only for increasing bone mineral density (BMD) but also improving physical fitness than treatment with BP alone [6].

The association between PPI use and the risk of fractures (hip, wrist or spine) is known to be stronger when the PPI is used at high doses or over a long-term period [17]. However, studies about the interaction between BP and PPIs and the risk of fracture are based on the analysis of data over a short observation period; therefore, the studies lack information on long-term users. Therefore, we need to reconfirm the influence of the interaction between BP and PPIs on fracture risk in long-term PPI users.

The purpose of this study was to reconfirm the association between fracture risks according to the PPI usage period and to reconfirm the interaction between BP and PPIs in long-term users based on the Korean National Health Insurance database. The results of this study could be used by clinicians to prescribe safer and more effective treatments for gastrointestinal ailments in elderly patients with a history of long-term PPI usage, especially women aged 65 and older who are at high risk of fractures.

## Materials and methods

### Data source and ethical considerations

Data from the period January 2007 to December 2017 from the Korean National Health Insurance Sharing Service (NIHSS) database were used. The Korean NIHSS system includes the entire national population (~50 million people), and the database was established for claim reimbursements. Data on subject characteristics, clinical information, socioeconomic level of the beneficiary, and death records were included in the database. Clinical information including disease diagnosis codes based on the International Codes of Disease 10th Edition (ICD-10) Clinical Modification, treatments based on drug prescriptions, and health care costs were recorded.

Patients were not directly involved in the research, and only the secondary electronic database was used for the analysis. Informed consent was not required as the database maintained de-identification and anonymity of sampled individuals. This study was approved by the Cha University Institutional Review Board (protocol ID: 1044308-201703-HR021-01).

### Study design and selection of cases and controls

A nested case-control design was applied to this study. The study population included elderly women aged ≥65 years with claims for peptic ulcer or GERD (ICD 10 code: K21, K25–28) from January 2010 to June 2017. We excluded patients who had a claim for any cancer (ICD 10 code: C) or Paget's disease (ICD 10 code: M88) during the study period (from 2007 to 2017).

The study subjects were grouped into cases and controls. Those who had sustained at least one osteoporotic fracture were classified as cases.

An osteoporotic fracture was defined as a diagnosis of osteoporosis (ICD10 code: M81, M82) before the fracture or within three months of the fracture (wrist [ICD10 code: S422, S423, S525, S526], spine [ICD10 code: 10 code: M484, M485, S220, S221], hip [ICD10 code: S720, S721, S722]) or osteoporosis with a current pathological fracture (ICD10 code: M80). For controls who had no history of osteoporotic fracture, each control was assigned the same event date as the fracture event date of the corresponding matched case according to the index date (date of the first diagnosis of gastrointestinal disorders) and age at the event date. The observation period was set for three years prior to the event date. Patients who sustained any fracture including an osteoporotic fracture during the observation period were excluded from participating as a case or a preliminary control subject. After excluding patients with a fracture history, the final controls were selected through 1:2 matching of cases to controls on the basis of the presence of osteoporosis (ICD10 code: M80, 81, 82) and the Charlson comorbidity index (CCI) during the period 1 year prior to the event date.

### Exposure assessment

A PPI user was defined as a patient who received at least one PPI prescription during the observation period. PPIs considered in this study were those containing any of the seven ingredients (omeprazole, lansoprazole, dexlansoprazole, esomeprazole, pantoprazole, rabeprazole, ilaprazole) listed in the Korean National Health Insurance Formulary.

In order to compare the fracture OR for the PPI users according to the duration of exposure to PPI, we defined the duration of exposure to PPI as the total number of PPI prescription days during the 3-year observation period and divided it into five quintiles: less than 1 month (< 30 days), 1–3 months (30–89 days), 3–6 months (90–179 days), 6–12 months (180–364 days), and 1 year or more (≥ 365 days). Since fractures that occurred more than 1 year after

the last exposure to PPI can hardly be associated with the exposure to PPI, a sub-analysis was carried out on the patients who had sustained fractures within 12 months from the last day of PPI medication use after excluding the patients who had not used PPIs for 1 year or more prior to the fracture.

### Statistical analysis

As for the characteristics of the cases and controls, categorical variables were presented as frequency and percentage and continuous variables (age, CCI) as mean and standard deviation (SD). In order to determine whether the characteristics of the cases and controls were significantly different, statistical analysis was performed using a student's t-test or chi-squared test, as appropriate.

For each patient, comorbidities that are known risk factors for fracture were evaluated based on the ICD-10 codes indicated in any claim made within 12 months of the event date. The evaluated diseases were: rheumatoid arthritis (M05, M06, M45), hyperthyroidism (E05), chronic kidney disease (N18), chronic obstructive pulmonary disease (J44, J45), hypopituitarism (E23.0), hyperparathyroidism (E21), Cushing's syndrome (E24), vitamin D deficiency (E55.9), idiopathic hypercalciuria (E83.5), diabetes (E11-E14), hypertension (I10-I13, I15), chronic liver disease (K72.1, K73, K74), systemic lupus erythematosus (M32), inflammatory bowel disease (K50, K51), and osteoporosis (M80, M81). Concomitant medications included in the prescriptions issued 1 year before the event date were evaluated. The evaluated medications, as the risk factors for fracture were as follows: antiplatelets, non-steroidal anti-inflammatory drugs, glucocorticoids, anticonvulsants, anticoagulants, selective serotonin reuptake inhibitors, benzodiazepines, and tricyclic antidepressants. Bisphosphonate (BP), hormone replacement therapy, and other anti-osteoporotic medications were considered preventive factors of fractures. As patient lifestyle variables, we evaluated alcohol use, smoking, physical exercise, and body mass index (BMI) using the health checkup data issued closest to the event date. The influence of the interaction between BP and PPIs on fracture risk was evaluated by analyzing the OR for fracture according to the use of BP. To this end, we defined a case with at least one prescription of BP during the observation period as a BP-user and a case with no history of prescription as a BP non-user.

Conditional logistic regression was performed to determine the association between PPI use and fracture risk. Results were presented as an aOR and 95% confidence interval (CI). The interaction between PPIs and BP was determined by calculating the aOR after dividing the dataset into BP user and non-user groups. All data were analyzed statistically using SAS version 9.3 (SAS Institute Inc., Cary, NC, USA), and statistical significance was set at $p < 0.05$ for a two-sided test.

### Results

A total of 151,155 women aged 65 years and over were diagnosed with peptic ulcer disease (PUD) or GERD between January 2010 and June 2017. Of them, 21,754 patients were selected as cases of osteoporotic fracture and 43,508 patients as controls with no history of fracture after the preliminary matching based on age and final date and the final matching based on the diagnosis of osteoporosis and CCI score (Fig 1).

Cases and controls were well-matched in terms of mean age (74 years), CCI score (1.83–1.85), and diagnosis of osteoporosis (32–34%). The controls showed a higher proportion in the 1st–3rd income quintiles, and the cases in the 4th and 5th income quintiles. A higher proportion of the cases lived in urban areas, and a higher proportion of the controls lived in metropolitan areas. The cases had higher comorbidity rates than the controls with regard to the

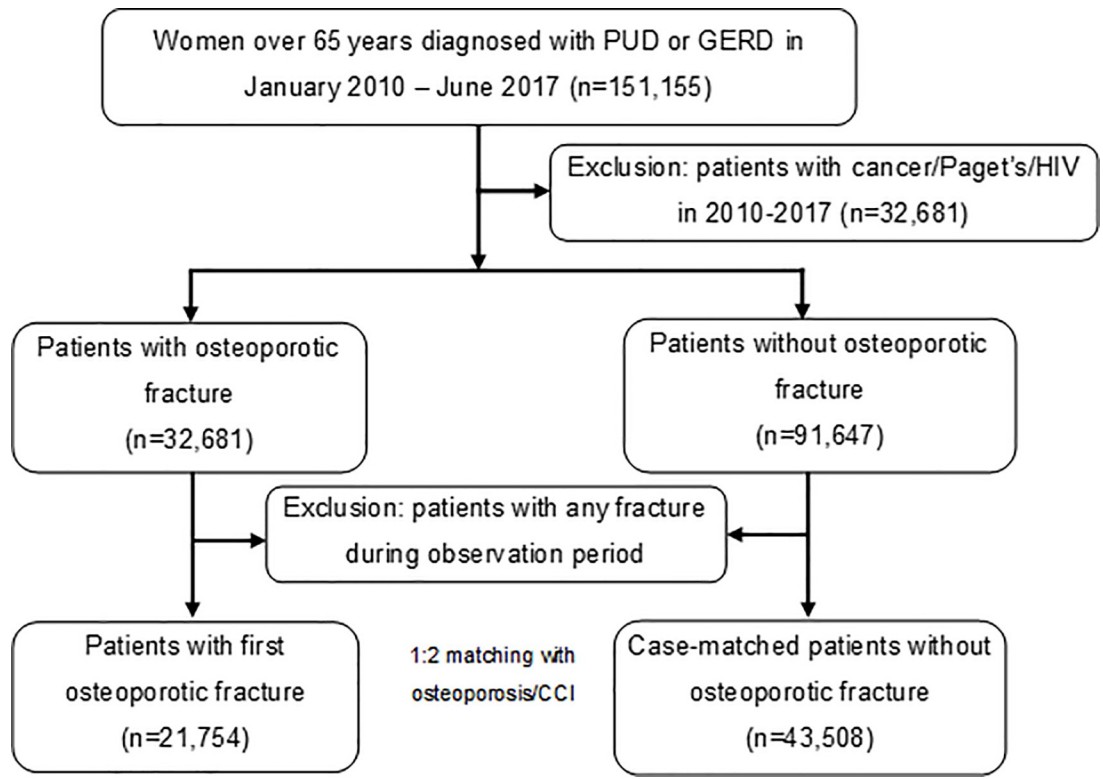

**Fig 1. Flowchart of case and control selection.** PUD: Peptic ulcer disease, GERD: gastroesophageal reflux disease, HIV: human immunodeficiency virus; CCI, Charlson comorbidity index.

diseases considered to be risk factors for fracture. The proportion of patients with concomitant medications known to increase fracture risk was also higher in the cases, as was the proportion of patients receiving anti-osteoporotic treatments other than hormone replacement therapy (Table 1).

The overall crude OR for fractures according to PPI use was 1.39 (95% CI: 1.35–1.44), which indicated a statistically significant association between PPI use and fracture risk. Even after adjusting for comorbidities and concomitant medications, and the aOR stood at 1.15 (95% CI: 1.11–1.20), demonstrating that the association remained statistically significant (Table 2).

Analysis performed separately on the BP user and non-user groups revealed the OR for fracture among BP users to be higher than that among non-users [aOR: 1.15 (95% CI: 1.08–1.92), vs. 1.11, (1.03–1.20)], demonstrating a statistically significant interaction between PPIs and BP (p<0.01). In both BP user and non-user groups, the OR for fracture increased as the duration of PPI use increased, showing that there is a positive dose-response relationship between PPI use and risk of osteoporotic fracture.

The sub-group analysis performed on the patients exposed to PPI within 12 months prior to fracture showed a higher OR compared with the base analysis (aOR 1.19 vs. 1.15), whereby all ORs increased regardless of the duration of exposure or BP use (Table 3).

## Discussion

In this nested case-control study, we investigated the association between the use of PPIs and the risk of osteoporotic fracture in elderly women (≥65 years) at high risk of osteoporotic

**Table 1. Demographic and clinical information for cases and controls.**

| | | Cases | | Controls | | p-value |
|---|---|---|---|---|---|---|
| | | (n = 21,754) | | (n = 43,508) | | |
| | | n | % | N | % | |
| Age, years | Mean (SD) | 74.21 (6.23) | | 73.93 (6.88) | | Matched |
| | Median | 73 | | 73 | | |
| Income level | 1st quintile | 2,883 | 13.25 | 6,753 | 15.52 | <0.001 |
| | 2nd quintile | 1,900 | 8.73 | 4,249 | 9.77 | |
| | 3rd quintile | 2,504 | 11.51 | 5,323 | 12.23 | |
| | 4th quintile | 3,835 | 17.63 | 7,230 | 16.62 | |
| | 5th quintile | 7,744 | 35.60 | 12,103 | 27.82 | |
| | unknown | 2,888 | 13.28 | 7,854 | 18.05 | |
| Residence* area | Metropolitan | 3,455 | 15.88 | 8,030 | 18.46 | <0.001 |
| | Cities | 4,707 | 21.64 | 9,213 | 21.18 | |
| | Rural areas | 13,592 | 62.48 | 26,269 | 60.38 | |
| CCI** | Mean (SD) | 1.85 (1.94) | | 1.83 (1.83) | | Matched |
| | 0–2 | 14,917 | 68.57 | 30,639 | 70.42 | |
| | ≥3 | 6,837 | 31.43 | 12,873 | 29.59 | |
| Comorbidity** | Rheumatism | 2,135 | 9.81 | 2,232 | 5.13 | <0.001 |
| | Hyperthyroidism | 506 | 2.33 | 822 | 1.89 | 0.0002 |
| | Chronic kidney disease | 401 | 1.84 | 632 | 1.45 | 0.0002 |
| | COPD | 5,212 | 23.96 | 7,958 | 18.29 | <0.001 |
| | Hypopituitarism | 23 | 0.11 | 28 | 0.06 | 0.0745 |
| | Hyperparathyroidism | 68 | 0.31 | 85 | 0.20 | 0.0035 |
| | Cushing's syndrome | 74 | 0.34 | 42 | 0.10 | <0.001 |
| | Vitamin D deficiency | 488 | 2.24 | 536 | 1.23 | <0.001 |
| | Idiopathic hypercalciuria | 624 | 2.87 | 655 | 1.51 | <0.001 |
| | Diabetes | 7,659 | 35.21 | 13,182 | 30.30 | <0.001 |
| | Hypertension | 14,795 | 68.01 | 26,500 | 60.91 | <0.001 |
| | Chronic hepatic disease | 729 | 3.35 | 1,240 | 2.85 | 0.0004 |
| | SLE | 26 | 0.12 | 37 | 0.09 | 0.1811 |
| | IBD | 59 | 0.27 | 73 | 0.17 | 0.0056 |
| | Osteoporosis | 14,381 | 33.89 | 13,772 | 31.65 | Matched |
| Medication** | Antiplatelet | 8,148 | 37.46 | 14,139 | 32.50 | <0.001 |
| | NSAID | 20,164 | 92.69 | 32,571 | 74.86 | <0.001 |
| | Glucocorticoid | 10,593 | 48.69 | 16,362 | 37.61 | <0.001 |
| | Anticonvulsant | 3,271 | 15.04 | 3,998 | 9.19 | <0.001 |
| | Anticoagulant | 1,681 | 7.73 | 1,999 | 4.59 | <0.001 |
| | SSRI | 1,562 | 7.18 | 1,832 | 4.21 | <0.001 |
| | Benzodiazepine | 13,590 | 62.47 | 20,371 | 46.82 | <0.001 |
| | Tricyclic antidepressant | 2,404 | 11.05 | 2,970 | 6.83 | <0.001 |
| | Bisphosphonate | 10,493 | 48.23 | 7,008 | 16.11 | <0.001 |
| | HRT | 239 | 1.10 | 673 | 1.55 | <0.001 |
| | Other osteoporosis therapy | 1,524 | 7.01 | 956 | 2.20 | <0.001 |
| Smoking*** | Yes | 355 | 1.63 | 830 | 1.91 | <0.001 |
| | No | 15,704 | 72.19 | 29,721 | 68.31 | |
| | Unknown | 5,695 | 26.18 | 12,961 | 29.79 | |
| Alcohol*** | Over allowance | 62 | 0.29 | 125 | 0.29 | 0.8445 |
| | Under allowance | 5,357 | 24.63 | 10,805 | 24.83 | |

(*Continued*)

**Table 1.** (Continued)

|  |  | Cases | | Controls | | p-value |
|---|---|---|---|---|---|---|
|  |  | **(n = 21,754)** | | **(n = 43,508)** | |  |
|  |  | **n** | **%** | **N** | **%** |  |
|  | Unknown | 16,335 | 75.09 | 32,582 | 74.89 |  |
| Exercise*** | Yes | 2,942 | 13.52 | 6,137 | 14.11 | <0.001 |
|  | No | 5,906 | 27.15 | 9,806 | 22.54 |  |
|  | Unknown | 12,904 | 59.32 | 27,569 | 63.37 |  |
| BMI%*** | Mean (SD) | 23.86 (3.38) | | 24.29 (3.41) | | <0.001 |

SD: standard deviation, BMI: body mass index, CCI: Charlson's Comorbidity Index, COPD: chronic obstructive pulmonary disease, SLE: systemic lupus erythematosus, IBD: inflammatory bowel disease, SSRI: selective serotonin reuptake inhibitors, NSAID: nonsteroidal anti-inflammatory drug: HRT, hormone replacement therapy.

* Definitions of residential areas: Metropolitan: A local government with boroughs; Cities: A local government with a population of 50,000 or more without a borough; Rural area: A local government with a population less than 50,000

**During 1-year before event date

***Date closest to event date.

fracture. While 57.7% of cases used PPIs, 49.8% in the control group used PPIs, showing a statistically significant association between PPI use and fracture risk (aOR: 1.15, 95% CI: 1.11–1.20). The correlation analysis between fracture risk and the duration of PPI exposure showed

**Table 2.** Osteoporotic fracture risk related to proton pump inhibitor use according to the duration of exposure stratified by the use of bisphosphonate.

|  | Proton pump inhibitor use | Cases (n = 21,754) | | Control (n = 43,508) | | Crude* odds ratio | Adjusted** odds ratio |
|---|---|---|---|---|---|---|---|
|  |  | **n** | **%** | **n** | **%** |  |  |
| All | Unexposed | 9,201 | 42.3 | 21,858 | 50.2 | Reference |  |
|  | Exposed | 12,553 | 57.7 | 21,650 | 49.8 | 1.39 (1.35, 1.44) | 1.15 (1.11, 1.20) |
|  | <30 days | 5,012 | 23.0 | 10,196 | 23.4 | 1.18 (1.13, 1.23) | 1.03 (0.98, 1.08) |
|  | ≤30 days, <90 days | 3,282 | 15.1 | 5,712 | 13.1 | 1.39 (1.32, 1.46) | 1.12 (1.06, 1.19) |
|  | ≤90 days, <180 days | 1,554 | 7.1 | 2,417 | 5.6 | 1.57 (1.46, 1.68) | 1.23 (1.14, 1.34) |
|  | ≤180 days, <365 days | 1,237 | 5.7 | 1,717 | 3.9 | 1.76 (1.63, 1.90) | 1.35 (1.23, 1.49) |
|  | ≤365 days | 1,468 | 6.7 | 1,608 | 3.7 | 2.24 (2.08, 2.42) | 1.72 (1.57, 1.89) |
| BP non-users | Unexposed | 5,075 | 40.0 | 5,515 | 43.7 | Reference |  |
|  | Exposed | 7,597 | 60.0 | 7,093 | 56.3 | 1.22 (1.14, 1.30) | 1.11 (1.03, 1.20) |
|  | <30 days | 2,916 | 23.0 | 3,008 | 23.9 | 1.07 (0.98, 1.16) | 1.01 (0.92, 1.11) |
|  | ≤30 days, <90 days | 1,976 | 15.6 | 1,955 | 15. | 1.18 (1.07, 1.30) | 1.06 (0.95, 1.18) |
|  | ≤90 days, <180 days | 973 | 7.7 | 904 | 7.2 | 1.19 (1.05, 1.36) | 1.05 (0.91, 1.21) |
|  | ≤180 days, <365 days | 799 | 6.3 | 623 | 4.9 | 1.54 (1.34, 1.78) | 1.36 (1.16, 1.60) |
|  | ≤365 days | 933 | 7.4 | 603 | 4.8 | 1.86 (1.61, 2.14) | 1.64 (1.40, 1.92) |
| BP Users | Unexposed | 4,126 | 45.4 | 16,343 | 52.9 | Reference |  |
|  | Exposed | 4,956 | 54.6 | 14,557 | 47.1 | 1.36 (1.29, 1.44) | 1.15 (1.08, 1.22) |
|  | <30 days | 2,096 | 23.1 | 7,188 | 23.3 | 1.14 (1.06, 1.22) | 0.99 (0.92, 1.07) |
|  | ≤30 days, <90 days | 1,306 | 14.4 | 3,757 | 12.2 | 1.49 (1.36, 1.62) | 1.22 (1.10, 1.34) |
|  | ≤90 days, <180 days | 581 | 6.4 | 1,513 | 4.9 | 1.52 (1.35, 1.72) | 1.27 (1.11, 1.46) |
|  | ≤180 days, <365 days | 438 | 4.8 | 1,094 | 3.5 | 1.68 (1.46, 1.93) | 1.33 (1.13, 1.55) |
|  | ≤365 days | 535 | 5.9 | 1,005 | 3.3 | 2.12 (1.85, 2.43) | 1.79 (1.53, 2.09) |

BP: bisphosphonate.

* Calculated by conditional logistic regression.

** Calculated by conditional regression adjusted for Charlson's comorbidity index, comorbidity, and medication.

**Table 3. Osteoporotic fracture risk among patients exposed to proton pump inhibitor within 1 year prior to fracture date.**

| | Proton pump inhibitor use | Cases (n = 17,902) | | Control (n = 35,665) | | Crude* odds ratio | Adjusted** odds ratio |
|---|---|---|---|---|---|---|---|
| | | **n** | **%** | **n** | **%** | | |
| All | Unexposed | 9,201 | 51.4 | 21,858 | 61.3 | Reference | |
| | Exposed | 8,701 | 48.6 | 13,807 | 38.7 | 1.54 (1.48, 1.60) | 1.19 (1.13, 1.24) |
| | <30 days | 2,840 | 15.9 | 5,486 | 15.4 | 1.26 (1.19, 1.33) | 1.02 (0.96, 1.09) |
| | ≤30 days, <90 days | 2,187 | 12.2 | 3,622 | 10.2 | 1.49 (1.40, 1.59) | 1.11 (1.03, 1.20) |
| | ≤90 days, <180 days | 1,212 | 6.8 | 1,768 | 5.0 | 1.73 (1.59, 1.88) | 1.30 (1.17, 1.44) |
| | ≤180 days, <365 days | 1,067 | 6.0 | 1,416 | 4.0 | 1.84 (1.68, 2.01) | 1.36 (1.22, 1.52) |
| | ≤365 days | 1,395 | 7.8 | 1,515 | 4.2 | 2.24 (2.06, 2.44) | 1.71 (1.55, 1.89) |
| BP non-users | Unexposed | 5,075 | 48.8 | 5,515 | 53.9 | Reference | |
| | Exposed | 5,319 | 51.2 | 4,710 | 46.1 | 1.31 (1.21, 1.42) | 1.13 (1.03, 1.23) |
| | <30 days | 1,649 | 15.9 | 1,651 | 16.1 | 1.11 (0.99, 1.24) | 0.99 (0.87, 1.12) |
| | ≤30 days, <90 days | 1,333 | 12.8 | 1,273 | 12.4 | 1.19 (1.05, 1.35) | 1.00 (0.87, 1.15) |
| | ≤90 days, <180 days | 766 | 7.4 | 702 | 6.9 | 1.35 (1.15, 1.57) | 1.15 (0.96, 1.37) |
| | ≤180 days, <365 days | 682 | 6.6 | 514 | 5.0 | 1.65 (1.39, 1.95) | 1.39 (1.15, 1.68) |
| | ≤365 days | 889 | 8.6 | 570 | 5.6 | 1.84 (1.57, 2.15) | 1.60 (1.34, 1.92) |
| BP users | Unexposed | 4,126 | 55.0 | 16,343 | 64.2 | Reference | |
| | Exposed | 3,382 | 45.0 | 9,097 | 35.8 | 1.50 (1.40, 1.60) | 1.21 (1.12, 1.30) |
| | <30 days | 1,191 | 15.9 | 3,835 | 15.1 | 1.21 (1.11, 1.33) | 1.03 (0.93, 1.14) |
| | ≤30 days, <90 days | 854 | 11.4 | 2,349 | 9.2 | 1.64 (1.47, 1.83) | 1.27 (1.12, 1.44) |
| | ≤90 days, <180 days | 446 | 5.9 | 1,044 | 4.1 | 1.67 (1.43, 1.95) | 1.31 (1.10, 1.56) |
| | ≤180 days, <365 days | 385 | 5.1 | 902 | 3.5 | 1.67 (1.42, 1.97) | 1.25 (1.04, 1.50) |
| | ≤365 days | 506 | 6.7 | 945 | 3.7 | 2.14 (1.84, 2.49) | 1.73 (1.46, 2.06) |

BP: bisphosphonate.

* Calculated by conditional logistic regression.

** Calculated by conditional regression adjusted for Charlson's comorbidity index, comorbidity, and medication.

that the risk of osteoporotic fracture increased with an increase in the duration of PPI expo-sure. This tendency was pronounced in the BP user group (aOR: 1.15, 95% CI: 1.08–1.22), which had a higher OR than the BP non-user group (aOR: 1.11, 95% CI: 1.03–.20); thus, verify-ing the interaction between PPIs and BP and the influence on fracture risk.

The finding of this study that PPI use increases the risk of fracture is consistent with that of recent meta-analyses performed by Eom et al. based on the papers published up to 2010. They reported the OR for fracture associated with PPI use to be 1.29 (95% CI: 1.18–1.41) [17]. Another meta-analysis performed on observational studies published up until February 2015 reported a moderate association between femoral and vertebral fractures and PPI use (relative risk of femoral fracture: 1.26; 95% CI: 1.16–1.36, relative risk of vertebral fracture: 1.58; 95% CI: 1.3–1.82) [18]. The most recent systematic review and meta-analysis of the studies pub-lished up until February 2018 also reported an increased risk of fracture associated with PPI use (effect size: 1.28; 95% CI: 1.22–1.35). However, with regard to the association between the duration of PPI use and increased fracture risk, researchers have reported differing results. Whilst the meta-analysis conducted by Eom and Zhou reported that PPI use for more than 1 year and less than 1 year showed similar risk levels, Nassar and Richter [19] noted that fracture risk did increase with an increase in the duration of PPI use. Such discrepancies may be ascrib-able to the limitation associated with calculating the accurate number of days of drug exposure due to different calculation methods inherent in observational studies. We must also consider that PPIs are not taken continuously or taken for a long time, but are administered when

needed to patients with PUD or GERD, which are major indications for PPIs, for a finite period of time and then discontinued. If only the first exposed event from the index date is considered, or if the length of the observation period varies for each subject, the cumulative impact of PPI may be underestimated. In this study, the number of days of drug exposure was calculated by summing the PPI prescription days during the observation period (3 years prior to the fracture event date), regardless of whether PPI use was continuous. Assuming that increase in the risk of fractures with PPI is related to bone metabolism [20–22], recovery of the weakened bones is difficult; further, discontinuation of PPI does not immediately increase bone strength. Thus, it is reasonable to evaluate the fracture risk by calculating the cumulative exposure period of PPI by summing the cumulative PPI prescription days during the constant pre-defined observation period, and the positive dose response relationship shown in this study supports the causality between PPI use and fracture risk.

When comparing the results of this study with those of the study conducted by Lee et al. using the Korean National Health Insurance claim database, both showed that the use of PPIs increased the risk of fracture, but the aOR of this study was much lower (1.34 vs. 1.15) [16]. This may be explained by the fact that the mean age of the subjects in the study conducted by Lee et al. was higher than that in this study (77 vs. 74 years) and the overall study period was shorter (1.5 vs. 3 years). In other words, the association between PPI use and fracture incidence may have been overestimated because the period between the exposure and the incidence of fracture was relatively short and the corresponding patients were elderly. The analysis data used in Lee et al.'s study covers the years 2005 and 2006. It should be kept in mind that PPI use increased rapidly with the expansion of its coverage by insurance in 2008 [3], and therefore, the pattern of PPI use might have been different compared to PPI use patterns after 2010, which was the study period of our study. Furthermore, unlike this study, which only included PUD and GERD patients in its analysis, Lee et al.'s study did not set such inclusion criteria, and the clinical characteristics of the PPI user and non-user groups may have influenced the fracture incidence. Considering that PPI mediates risk factors, thus increasing fracture risk, the PPI prescription practice and the characteristics of PPI users can influence the study outcome. For instance, when rigorous insurance coverage criteria were applied to PPIs, PPIs were most likely administered to patients with severe gastrointestinal disorders who might have ingested a different quality of meals compared to patients without gastrointestinal disorders, and factors such as gastrointestinal malabsorption might have influenced their fracture risk. Therefore, this study was designed as a cohort study composed of gastrointestinal patients in order to minimize the effects of fracture on the patients' gastrointestinal disorders.

Although the mechanism by which PPIs increase fracture risk is yet to be determined, researchers have suggested that the increase in gastric acidity caused by PPI intake may adversely affect calcium absorption [20] and that the suppression of vacuolar $H^+$-ATPase in bone inhibits bone resorption, thus increasing fracture risk [21, 22]. However, the association between PPI use and decrease in bone density has yet to be property elucidated [23, 24]. There may be other viable mechanisms behind the PPI-induced fracture risk other than impaired bone structure. Despite research findings that PPI use is associated with hyperparathyroidism and hypocalcemia, it is still unclear whether PPI use induces hypocalcemia and secondary hyperparathyroidism or whether gastrointestinal disorders caused by hyperparathyroidism or hypocalcemia increase PPI use [25].

BP is a drug used for the treatment of osteoporosis, and it works by increasing bone density by suppressing osteoclasts [26]. PPIs are used widely to prevent or treat upper gastrointestinal disorders, one of the major side effects of BP [13]. BP adherence has also been reported as an interactive mechanism for increasing the PPI-induced fracture risk in BP users. BP users are likely to sustain gastrointestinal disorders as a side effect of BP, which may result in a low BP

adherence and thus decrease the anti-fracture effect of BP [13]. However, the study conducted by Lee et al. that demonstrated the interaction between BP and PPIs, no difference in BP adherence was observed in the PPI exposure and non-exposure groups among the BP users [16]. Both BP and PPIs suppress osteoclasts, and their interactive mechanism is explained by chronically impaired bone remodeling, which makes bone prone to fracture, while maintaining bone density. The usual dose of PPIs for treating gastrointestinal disorders is known to be too low to reach a blood concentration high enough to affect osteoclasts, which makes it implausible to explain the mechanism with PPIs alone. However, the pharmacological interactions of BP and PPIs administered together at their respective usual clinical doses have yet to be clarified through further research.

Unlike other studies conducted in Korea, this study used data generated after the rapid increase in the use of PPIs, and thus the derived analysis results reflect the characteristics of current PPI users. Another advantage of this study was its observation period of 3 years, which is neither too long to prevent the dilution of the association between PPI use and fracture risk nor too short to observe the effects of long-term PPI use.

This study had some limitations. Given the nature of health insurance claim data, there are no detailed descriptions of the patients' clinical features, so our analysis was based only on the patient information obtained from the disease codes required for insurance claims. Although some clinical parameters, such as BMD, are closely related to the risk of fracture [27], the database used in this study could not provide this information and thus did not reflect it. Since osteoporosis, which had a significant effect on fractures in this study, is a disease with few external symptoms, it is likely that a large number of patients without a diagnosis of osteoporosis were included in the study population. In fact, cases included in this study were defined as patients with osteoporotic fractures; however, approximately 34% of these cases had been diagnosed with osteoporosis before the fracture occurred, and 67% were diagnosed with osteoporosis based on active diagnostic tests following the fracture. Thus, we can assume that the control group may have included undiagnosed osteoporosis patients. Since PPI use is known to increase the risk of fractures, the diagnosis of osteoporosis is a factor that may affect PPI prescription. In other words, the prescription of PPIs for the treatment of gastrointestinal diseases may be avoided in patients diagnosed with osteoporosis. Therefore, in this study, the diagnosis of osteoporosis before the event date is considered a factor that may influence the drug selection of patients. We tried to minimize the effects of misclassification related to osteoporosis by using the diagnosis of osteoporosis before the event date as a matching variable.

Although life style variables, such as smoking, alcohol, physical activity, and body weight are important factors affecting BMD [27], this study did not use them as covariates. As the lifestyle information presented in Table 1 was obtained from the national health examination database, the time of measurement could not be specified, and the "unknown" ratio was more than half. Hence, it seemed inaccurate to include these parameters as covariates. Smoking or obesity are risk factors for fractures as well as gastric ulcers or GERD [28]. Smokers and obese people are more likely to use PPI to treat gastrointestinal problems, and the case group might have had a higher proportion of smokers and obese people and PPI users. However, this study performed in the nested cohort included patients with peptic ulcer or GERD. As all subjects had gastrointestinal problems, we minimized the impact of the mechanism described above as a confounding factor in the findings.

Furthermore, we could not account for factors such as the use of calcium supplements due to a lack of available information on dietary supplements or over-the-counter drugs. Moreover, due to the nature of a case-control study, it was impossible to demonstrate a direct causal relationship between PPI use and fracture risk and the underlying mechanisms. In the use of PPIs and BP, the temporal and sequential relationships could not be reflected in the analysis

given that there were many limitations in inferring the mechanisms by which their interactions affect fracture incidence. Therefore, care should be taken in interpreting the results of this study.

Nevertheless, this study has some strengths. We considered issues related to time in case-control matching to minimize the time-related bias (time-window bias) that is likely to occur in case-control studies [29]. The same event date was assigned to the case and matched controls, and the same observation period of 3 years was applied from the event date for both groups. Thus, this study design eliminated the tendency of over-representation of unexposed cases, which could be caused by differences in the length of observation period between case and control.

In conclusion, the results of this study showed that care should be taken when administering PPIs to elderly women at high risk of fracture, especially in cases of long-term medication use. Particular caution is warranted when prescribing BP for patients who are also using PPIs.

## Acknowledgments

This study analyzed NIHSS data (NHIS-2018-1-391) provided by the Korean National Health Insurance Service (NHIS). The authors declare no potential conflicts of interest with NHIS with respect to the authorship and/or publication of this article.

## Author Contributions

**Conceptualization:** Jong Joo Kim, Eun Jin Jang, Jiwon Park, Hyun Soon Sohn.

**Data curation:** Jong Joo Kim, Eun Jin Jang, Jiwon Park.

**Formal analysis:** Eun Jin Jang, Jiwon Park.

**Funding acquisition:** Hyun Soon Sohn.

**Investigation:** Jong Joo Kim, Eun Jin Jang, Hyun Soon Sohn.

**Methodology:** Jong Joo Kim, Eun Jin Jang, Jiwon Park, Hyun Soon Sohn.

**Project administration:** Jong Joo Kim.

**Resources:** Jong Joo Kim, Eun Jin Jang, Hyun Soon Sohn.

**Software:** Eun Jin Jang, Jiwon Park.

**Supervision:** Hyun Soon Sohn.

**Validation:** Jong Joo Kim, Eun Jin Jang, Jiwon Park, Hyun Soon Sohn.

**Visualization:** Jong Joo Kim.

**Writing – original draft:** Jong Joo Kim, Eun Jin Jang, Jiwon Park.

**Writing – review & editing:** Jong Joo Kim, Eun Jin Jang, Jiwon Park, Hyun Soon Sohn.

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
