## [Decision Letter · Decision Letter 0]

3 Dec 2019

PONE-D-19-30609

Association between proton pump inhibitor use and risk of fracture: a population-based case-control study.

PLOS ONE

Dear Mrs Sohn,

Thank you for submitting your manuscript to PLOS ONE. After careful consideration, we feel that it has merit but does not fully meet PLOS ONE’s publication criteria as it currently stands. Therefore, we invite you to submit a revised version of the manuscript that addresses the points raised during the review process.

Both reviewers have indicated the need to characterize the study population more completely.  In particular, additional details regarding fractures and prior osteoporosis diagnosis are necessary.  

We would appreciate receiving your revised manuscript by Jan 17 2020 11:59PM. To enhance the reproducibility of your results, we recommend that if applicable you deposit your laboratory protocols in protocols.io, where a protocol can be assigned its own identifier (DOI) such that it can be cited independently in the future. For instructions see: http://journals.plos.org/plosone/s/submission-guidelines#loc-laboratory-protocols

We look forward to receiving your revised manuscript.

Kind regards,

Robert Daniel Blank, MD, PhD

Academic Editor

PLOS ONE

Journal Requirements:

2. Please amend either the title on the online submission form (via Edit Submission) or the title in the manuscript so that they are identical.

Additional Editor Comments (if provided):

Reviewers' comments:

Reviewer's Responses to Questions

**Comments to the Author**

1. Is the manuscript technically sound, and do the data support the conclusions?

Reviewer #1: Partly

Reviewer #2: Yes

2. Has the statistical analysis been performed appropriately and rigorously? 

Reviewer #1: No

Reviewer #2: Yes

3. Have the authors made all data underlying the findings in their manuscript fully available?

Reviewer #1: Yes

Reviewer #2: Yes

4. Is the manuscript presented in an intelligible fashion and written in standard English?

Reviewer #1: Yes

Reviewer #2: Yes

5. Review Comments to the Author

Reviewer #1: The authors conducted a nested case-control study to reconfirm (i) the association between the proton-pump inhibitors (PPI) usage and fracture risks, and (ii) the interaction between bisphosphonate (BP) and PPI in long-term users. The Korean National Health Insurance Sharing Service database covering the period Jan 2007 to Dec 2017 was used. The study subjects who had sustained at least one osteoporotic fracture, operationally defined as a “diagnosis of osteoporosis prior to or within three months of the fracture diagnosis or osteoporosis with current pathological fracture” were identified as cases. It is not clear how fracture was identified from the database, and whether all or only three specific fractures (i.e. “wrist, spine, femur”) were examined in the study. Two controls who had no history of osteoporotic fracture were matched by age, the presence of osteoporosis and the Charlson comorbidity index. The study subjects who had had any fracture within three years prior to the event date (i.e. the fracture date for cases) were excluded from the analyses. The conditional logistic regression analyses, additionally accounting for Charlson index, comorbidities and medications revealed an increased odds of any fracture associated with PPI (aOR: 1.15; 95% CI: 1.11, 1.20). Interestingly, the study also reconfirmed the interaction between BP and PPI in long-term users regardless of the trivial difference (1.15 (95% CI: 1.08, 1.92) in BP users vs. 1.11 (1.03, 1.20) in BP non-users).

The manuscript is overall informative. However, the authors might wish to address the following issues in the subsequent submission.

Major issues:

1. Study rationale

Several meta-analyses, such as Zhou B et al (cited as reference 18 in the manuscript) or Nassar Y et al (reference 19) have indicated the association between PPI and fracture risk, including hip fracture risk. The long-term PPI therapy and hip fracture risk was also investigated in a large nested case-control study of 13,556 hip fracture cases and 135,386 controls (i.e. Yang X et al, reference 20). Given the current literature, the authors are expected to make their study rationales stronger and much more convincingly. Speculation of how the findings would be used in clinical practice would be also useful to justify the study rationale.

2. Osteoporosis diagnosis and definition of osteoporotic fracture

It appears that diagnosis of osteoporosis is prerequisite to operationally define an osteoporotic fracture (“An osteoporotic fracture was defined as a diagnosis of osteoporosis prior to or within three months of the fracture (wrist, spine, femur) or osteoporosis with current pathological fracture”). However, approximately 66% of the cases had osteoporosis comorbidity reported (Table 1). Please explain how cases (i.e. the study subjects with an osteoporotic fracture) who did not have osteoporosis diagnosis were selected.

Similarly, the authors also acknowledged “a large number of patients without a diagnosis of osteoporosis were included in the control group”. Further clarification would be useful as the controls were matched by osteoporosis diagnosis known as a prerequisite condition for selecting cases.

3. Fracture types

I believe that the study might have aimed to examine all fracture types, but only three specific fractures (“wrist, spine, femur”) were mentioned in the manuscript. Importantly, it is not clear how a fracture was identified (i.e. using ICD 10 codes or self-report) and which fracture types were included.

Please verify how a fracture was identified in this study. If ICD 10 codes were used to capture a fracture, these ICD 10 codes should be provided, at least as an appendix table. If the study indeed examined only three specific fractures, please justify the reasons why these specific fractures were selected, how they were identified, and whether “femur” included hip fracture.

The authors might also wish to discuss how misclassification of osteoporotic fracture, such as a traumatic fracture reported in a patient with osteoporosis diagnosis was possible and how it might affect their findings.

4. Validation of diagnoses derived from this database

It would be useful if the authors could explain how accurate and reliable the diagnoses derived from reported ICD 10 codes. Are there published efforts to validate these diagnoses? If not, the authors should acknowledge and discuss potential impact of any misclassification on their findings.

5. Statistical analysis

In addition to the matching design, it is reasonable to make adjustment for several covariates such as Charlson comorbidity index, comorbidities and medication. The adjusted analyses actually attenuated the association between PPI and any fracture risk (from 1.39 (95% CI: 1.35, 1.44) to 1.15 (1.11, 1.20). Please make clear whether these covariates were predefined or data driven.

The authors are expected to justify why the analyses did not account for other known risk factors of fracture (such as smoking, physical activity and importantly, BMI as a proxy of BMD) that might have confounded the association between PPI and fracture risk. The authors might also wish to present the results of analyses accounting for all potential confounding effects, to at least confirm the robustness of the findings, even though such analyses were not predefined.

6. Possible impact of immortal time bias on the findings

The study also reported an increased fracture risk associated with increased duration of PPI use (Table 2). As the treatment was not randomly allocated, patients with longer duration of PPI would be possibly followed long enough to sustain a fracture, while those with short treatment duration might have died before they had a chance to sustain a fracture (i.e. an immortal time bias). Given its commonness in observational pharmaco-epidemiological studies, please discuss whether and how this immortal bias would impact the findings.

Minor issues:

- Categorization of duration of PPI was purely data driven. Are other more clinically relevant categorizations available? If so, the authors might wish to consider a sensitivity analysis using more clinically relevant categorisation of PPI duration.

Reviewer #2: This is a nested case-control epidemiological study of elderly females with a diagnosis of peptic ulcer disease or GERD from a Korean National Health Insurance Sharing Service. The results confirm previous studies and meta-analyses that indicate a significant association between use of proton pump inhibitors and osteoporotic fractures. In addition, a significant interaction was found between PPI and bisphosphonate use and fracture risk. The study is well done with clearly-stated results.

1. The authors required a diagnosis of osteoporosis prior to or within 3 months of a fracture for the case group. Please specify whether the diagnosis of osteoporosis was made by bone density studies or by a clinical diagnosis.

2. Many fractures in elderly patients occur in the absence of osteoporosis. Would the results differ if all patients with fragility fractures were included in the case group regardless of whether they had an ICD-10 diagnosis of osteoporosis?

3. Please indicate how “urban” differs from “metropolitan” in this data base.

6. PLOS authors have the option to publish the peer review history of their article (what does this mean?). If published, this will include your full peer review and any attached files.

Reviewer #1: Yes: Thach Tran

Reviewer #2: No

---

## [Author Response · Author response to Decision Letter 0]

16 Apr 2020

Thanks for two reviewers’ kind comments on our paper. We did reflect the matters you pointed out as much as possible. Please find the uploaded file_response to reviewers.

Reviewer #1: 

Major issues:

1. Study rationale

Several meta-analyses, such as Zhou B et al (cited as reference 18 in the manuscript) or Nassar Y et al (reference 19) have indicated the association between PPI and fracture risk, including hip fracture risk. The long-term PPI therapy and hip fracture risk was also investigated in a large nested case-control study of 13,556 hip fracture cases and 135,386 controls (i.e. Yang X et al, reference 20). Given the current literature, the authors are expected to make their study rationales stronger and much more convincingly. Speculation of how the findings would be used in clinical practice would be also useful to justify the study rationale.

In accordance with the important points you gave, the clinical meaning has been further refined as follows: 

It is significant that the association between PPI use and the risk of fracture especially depending on bisphosphonate(BP) exposure in Korean medical environment different from previous studies, was confirmed in a similar pattern. The study results could be used as an evidence for clinicians to make a decision for more effective and safe treatment of gastrointestinal problems in elderly with long-term PPI usage, especially women aged 65 and older at high risk of fracture.

2. Osteoporosis diagnosis and definition of osteoporotic fracture

It appears that diagnosis of osteoporosis is prerequisite to operationally define an osteoporotic fracture (“An osteoporotic fracture was defined as a diagnosis of osteoporosis prior to or within three months of the fracture (wrist, spine, femur) or osteoporosis with current pathological fracture”). However, approximately 66% of the cases had osteoporosis comorbidity reported (Table 1). Please explain how cases (i.e. the study subjects with an osteoporotic fracture) who did not have osteoporosis diagnosis were selected.

Similarly, the authors also acknowledged “a large number of patients without a diagnosis of osteoporosis were included in the control group”. Further clarification would be useful as the controls were matched by osteoporosis diagnosis known as a prerequisite condition for selecting cases.

We applied different time periods for osteoporosis evaluation to select cases and for comorbidity assessment to explain patient characteristics. 

In order to select the case, subjects were evaluated for 3 months after the fracture and whole history period before the fracture. Comorbidities described in Table 1 were evaluated for 1 year before the fracture.

We used data about osteoporosis and CCI for 1-year before the fracture as a matching variable because osteoporosis diagnosis probably affect the choice of drug. 

These explanation have been added in the Discussion.

3. Fracture types

I believe that the study might have aimed to examine all fracture types, but only three specific fractures (“wrist, spine, femur”) were mentioned in the manuscript. Importantly, it is not clear how a fracture was identified (i.e. using ICD 10 codes or self-report) and which fracture types were included. 

Please verify how a fracture was identified in this study. If ICD 10 codes were used to capture a fracture, these ICD 10 codes should be provided, at least as an appendix table. If the study indeed examined only three specific fractures, please justify the reasons why these specific fractures were selected, how they were identified, and whether “femur” included hip fracture.

The authors might also wish to discuss how misclassification of osteoporotic fracture, such as a traumatic fracture reported in a patient with osteoporosis diagnosis was possible and how it might affect their findings.

ICD10 codes for fractures have been inserted in Method. 

Chief clinical manifestations of osteoporosis are vertebral and hip fractures, and wrist fracture are obviously affected by osteoporosis. Therefore, only three major fractures due to osteoporosis (spine wrist and hip), were defined as events interested in this study. 

As you pointed out, it is possible that some trauma fractures are included, but osteoporotic bone is more likely to fracture than is normal bone at any level of trauma. So a fracture with osteoporosis diagnosis defined as an osteoporotic fracture regardless of trauma. 

(ref. Robert Lindsay and Felicia Cosman. Chapter 404: osteoporosis. McGraw-Hill Education. Harrison’s principle of internal medicine. 20th ed.) 

4. Validation of diagnoses derived from this database

It would be useful if the authors could explain how accurate and reliable the diagnoses derived from reported ICD 10 codes. Are there published efforts to validate these diagnoses? If not, the authors should acknowledge and discuss potential impact of any misclassification on their findings.

The database used in this study covers the entire population with a single National Health Insurance in Korea, and it has the advantage of minimizing recall bias that is common in case-control studies. But it has not been validated for the accuracy and reliability of the diagnoses from the database, since it build up for insurance claims. 

In particular, in the case of osteoporosis, the diagnosis is often missed when there are no special events or symptoms, and as noted above, it is difficult to distinguish traumatic fractures. Please kindly refer to the above mentioned explanation and revised Discussion.

5. Statistical analysis

In addition to the matching design, it is reasonable to make adjustment for several covariates such as Charlson comorbidity index, comorbidities and medication. The adjusted analyses actually attenuated the association between PPI and any fracture risk (from 1.39 (95% CI: 1.35, 1.44) to 1.15 (1.11, 1.20). Please make clear whether these covariates were predefined or data driven.

The authors are expected to justify why the analyses did not account for other known risk factors of fracture (such as smoking, physical activity and importantly, BMI as a proxy of BMD) that might have confounded the association between PPI and fracture risk. The authors might also wish to present the results of analyses accounting for all potential confounding effects, to at least confirm the robustness of the findings, even though such analyses were not predefined.

All covariates are predefined as factors known to be fracture risk or preventive factors, based on previous research or literature. 

Variables related with lifestyle were not included as covariates in the analysis. Since the data originated from national health examination, the time of measurement could not be specified and the “unknown” ratio was more than half. So it seemed inaccurate to use as covariates. Smoking or obesity are risk factors for fractures, gastric ulcers or GERD. Smokers and obese people are more likely to use PPI to treat gastrointestinal problems, and the case group may have higher proportion of them and PPI users. However, this study performed in the nested cohort with patients with peptic ulcer or GERD. Since all subjects have gastrointestinal problems, the mechanism described above is unlikely to be a confounding factor in the findings.

We agree that BMD value is very important variable for fracture, but unfortunately the Korean National Health Insurance Sharing Service (NIHSS) database we used in this study does not include BMD value of the patients.

6. Possible impact of immortal time bias on the findings

The study also reported an increased fracture risk associated with increased duration of PPI use (Table 2). As the treatment was not randomly allocated, patients with longer duration of PPI would be possibly followed long enough to sustain a fracture, while those with short treatment duration might have died before they had a chance to sustain a fracture (i.e. an immortal time bias). Given its commonness in observational pharmaco-epidemiological studies, please discuss whether and how this immortal bias would impact the findings.

In the cohort study design, the immortal time bias may be an important consideration, but in this case-control study where we select a subject who has developed a disease and then assess his/her past history, the subject who died or censored follow up is not included in the observation. We think that there might be a little impact of the immortal time bias in this study.

Minor issues:

- Categorization of duration of PPI was purely data driven. Are other more clinically relevant categorizations available? If so, the authors might wish to consider a sensitivity analysis using more clinically relevant categorisation of PPI duration.

The classification of duration of PPI use was simply predefined with a reference to previous studies. We agreed to your suggestion for more clinically relevant categorization of PPI exposure duration.

But, the database we used in this study was allowed to access remotely for a limited time we proposed. 

Now the database had been closed, so an additional analysis to apply the new classification requires a lot of time and new process to acquire the data. We would like to consider what you have pointed out in the further study.

Reviewer #2: 

1. The authors required a diagnosis of osteoporosis prior to or within 3 months of a fracture for the case group. Please specify whether the diagnosis of osteoporosis was made by bone density studies or by a clinical diagnosis.

As per your comment, ICD 10 codes for fractures have been inserted.

2. Many fractures in elderly patients occur in the absence of osteoporosis. Would the results differ if all patients with fragility fractures were included in the case group regardless of whether they had an ICD-10 diagnosis of osteoporosis?

Last year, we had conducted a similar study for the case group regardless of whether they had an ICD-10 diagnosis of osteoporosis in adults aged 50 and more, and results were similar with this study. As we establish a hypothesis that PPI increases the fracture risk via osteoporosis, we limit the association between PPI and osteoporotic fracture among the population with high risk of osteoporosis in this study. 

3. Please indicate how “urban” differs from “metropolitan” in this data base.

Residential areas were classified according to the definition below and the definition has been added in the Table 1. 

Metropolitan: A local government with boroughs

Cities: A local government with a population of 50,000 or more without a borough

Rural area: A local government with a population less than 50,000

---

## [Decision Letter · Decision Letter 1]

30 Apr 2020

PONE-D-19-30609R1

Association between proton pump inhibitor use and risk of fracture: a population-based case-control study.

PLOS ONE

Dear Mrs Sohn,

Thank you for submitting your manuscript to PLOS ONE. After careful consideration, we feel that it has merit but does not fully meet PLOS ONE’s publication criteria as it currently stands. Therefore, we invite you to submit a revised version of the manuscript that addresses the points raised during the review process.

Please see editor comments regarding the priority among review issues. 

We would appreciate receiving your revised manuscript by Jun 14 2020 11:59PM. To enhance the reproducibility of your results, we recommend that if applicable you deposit your laboratory protocols in protocols.io, where a protocol can be assigned its own identifier (DOI) such that it can be cited independently in the future. For instructions see: http://journals.plos.org/plosone/s/submission-guidelines#loc-laboratory-protocols

We look forward to receiving your revised manuscript.

Kind regards,

Robert Daniel Blank, MD, PhD

Academic Editor

PLOS ONE

Additional Editor Comments (if provided):

Addressing the points raised by Dr. Tran will, in my opinion, improve your manuscript. In particular, additional discussion of the body mass and time bias issues will be welcome. With regard to the motivation for your work, Dr. Tran's comments speak to how much attention this work is likely to attract in the future. However, I do not believe that this must be addressed for the paper to be published.

Reviewers' comments:

Reviewer's Responses to Questions

**Comments to the Author**

1. If the authors have adequately addressed your comments raised in a previous round of review and you feel that this manuscript is now acceptable for publication, you may indicate that here to bypass the “Comments to the Author” section, enter your conflict of interest statement in the “Confidential to Editor” section, and submit your "Accept" recommendation.

Reviewer #1: (No Response)

Reviewer #2: All comments have been addressed

2. Is the manuscript technically sound, and do the data support the conclusions?

Reviewer #1: Yes

Reviewer #2: (No Response)

3. Has the statistical analysis been performed appropriately and rigorously? 

Reviewer #1: Yes

Reviewer #2: (No Response)

4. Have the authors made all data underlying the findings in their manuscript fully available?

Reviewer #1: Yes

Reviewer #2: (No Response)

5. Is the manuscript presented in an intelligible fashion and written in standard English?

Reviewer #1: Yes

Reviewer #2: (No Response)

6. Review Comments to the Author

Reviewer #1: I would like to thank the authors for their time and efforts to address my concerns. There are however several minor issues they might wish to address more convincingly.

1. Study rationale

The authors provided speculations on how the findings would be useful. However, I found “the studies lack information on long-term users. Therefore, we need to reconfirm the influence of the interaction between BP and PPIs on fracture risk in long-term PPI users” not convincing enough.

I understand that the authors aimed to “reconfirm” the previous findings. However, they are expected to make the novelties and rationales far clearer and more convincing. More specifically, please clarify how the current findings could contribute to the current science knowledge on top of the previous findings from several meta-analyses (references 18, 19) and a very large nested case-control study (reference 20).

2. I agree with what the authors explained about covariates in the analyses. However, I’d like to suggest them to add the explanation they made for weight and BMD into the manuscript, which would make their analyses more convincing.

3. Possible time-related biases

I am sorry that my concerns about time-related biases (which was wrongly written as an immortal bias) might have caused a misunderstanding. While I agree with the authors that an immortal bias might not be problematic in their case-control study, I think it is worth discussing further the potential contribution, if any of time-related biases, especially the time-window bias. It is important as the cases and controls were matched by age, osteoporosis diagnosis and Charlson index. It would be an important strength if the time-related biases were already accounted for.

Reviewer #2: (No Response)

7. PLOS authors have the option to publish the peer review history of their article (what does this mean?). If published, this will include your full peer review and any attached files.

Reviewer #1: Yes: Thach Tran

Reviewer #2: No

---

## [Author Response · Author response to Decision Letter 1]

21 May 2020

Thanks for reviewer#1's kind comments on our paper. We did reflect the matters you pointed out as much as possible. 

1. Study rationale : The authors provided speculations on how the findings would be useful. However, I found “the studies lack information on long-term users. Therefore, we need to reconfirm the influence of the interaction between BP and PPIs on fracture risk in long-term PPI users” not convincing enough. I understand that the authors aimed to “reconfirm” the previous findings. However, they are expected to make the novelties and rationales far clearer and more convincing. More specifically, please clarify how the current findings could contribute to the current science knowledge on top of the previous findings from several meta-analyses (references 18, 19) and a very large nested case-control study (reference 20).

Author’s reply: Thank you for your comments regarding the study novelties. Basically we agreed with you because a novelty is important in academic researches. As we thought our study had the following strengths comparing to previous studies on the same topic. We have added three points separately in the Discussion.

There were discrepancies in the relationship between the duration of PPI use and fracture risk in the previous meta-analyses, due to applying different definitions of the PPI exposure duration in those studies. Considering the mechanism by which PPI use increases the risk of fracture, our study defined the PPI exposure duration as the cumulative prescription days for 3 years for each subject. With the strength of such research design, we could identify more robust association between PPI use and fracture risk. [Revised manuscript page 12, line 240~247]

Considering that PPI is the major drug indicated for GERD which affect fractures, this study was conducted only for patients with GERD or gastric ulcer. We enrolled patients with diagnosis of GERD or gastric ulcer, and the association of PPI use and fracture risk was analyzed for the cohort. This approach probably minimized confounding effect related with both gastric problems and fractures [Revised manuscript page 15, line 314~323]

We applied the same 3-year observation period on PPI exposure before index date (or same date in matching control) to both cases and controls. 3-years is a comparatively long-term observation than the previous studies, and the same length of the observation period could make time-window bias minimize. [Revised manuscript page 15, line 332~337]

2. I agree with what the authors explained about covariates in the analyses. However, I’d like to suggest them to add the explanation they made for weight and BMD into the manuscript, which would make their analyses more convincing.

Author’s reply: Thank you. As per your suggestion, we have added the explanation in the Discussion. [Revised manuscript page 15, line 314~313]

3. Possible time-related biases : I am sorry that my concerns about time-related biases (which was wrongly written as an immortal bias) might have caused a misunderstanding. While I agree with the authors that an immortal bias might not be problematic in their case-control study, I think it is worth discussing further the potential contribution, if any of time-related biases, especially the time-window bias. It is important as the cases and controls were matched by age, osteoporosis diagnosis and Charlson index. It would be an important strength if the time-related biases were already accounted for.

Author’s reply: Thank you for pointing out what we missed. You are right. The same event date and the same observation period were assigned to both cases and controls in our study, which make the possibility of time-window bias minimize. Accordingly, we added this sentences in the Discussion, as the above mentioned in the Answer #1. [Revised manuscript page 15, line 332~337]

---

## [Decision Letter · Decision Letter 2]

10 Jun 2020

Association between proton pump inhibitor use and risk of fracture: a population-based case-control study.

PONE-D-19-30609R2

Dear Dr. Sohn,

We’re pleased to inform you that your manuscript has been judged scientifically suitable for publication and will be formally accepted for publication once it meets all outstanding technical requirements.

Kind regards,

Robert Daniel Blank, MD, PhD

Academic Editor

PLOS ONE

Additional Editor Comments (optional):

Reviewers' comments:

Reviewer's Responses to Questions

**Comments to the Author**

1. If the authors have adequately addressed your comments raised in a previous round of review and you feel that this manuscript is now acceptable for publication, you may indicate that here to bypass the “Comments to the Author” section, enter your conflict of interest statement in the “Confidential to Editor” section, and submit your "Accept" recommendation.

Reviewer #1: (No Response)

Reviewer #2: All comments have been addressed

2. Is the manuscript technically sound, and do the data support the conclusions?

Reviewer #1: Partly

Reviewer #2: Yes

3. Has the statistical analysis been performed appropriately and rigorously? 

Reviewer #1: Yes

Reviewer #2: Yes

4. Have the authors made all data underlying the findings in their manuscript fully available?

Reviewer #1: Yes

Reviewer #2: Yes

5. Is the manuscript presented in an intelligible fashion and written in standard English?

Reviewer #1: Yes

Reviewer #2: Yes

6. Review Comments to the Author

Reviewer #1: I’d like to thank the authors for their efforts to address my concerns. There are two minor issues the authors might wish to address to make the manuscript more convincing.

1. The authors has indicated the novel definition of PPI exposure “as the cumulative prescription days for 3 years for each subject”. The authors might wish to add a couple of sentences to clarify how this novel operational definition is better than the previous ones, making the statement “to identify more robust association between PPI use and fracture risk” justified.

The authors might wish to use the evidence related to biological mechanism (i.e. how the novel definition is more biologically rationale) or research methodology (i.e. how it is methodologically less biased), rather than an overall, somewhat vague statement.

2. Similarly, please give an example of a confounding effect that the study was successfully able to account for. Such an example would be also useful to make the statement “This approach probably minimized confounding effect related with both gastric problems and fractures” clearer and more convincing.

Looking forward to receiving your response soon.

Reviewer #2: (No Response)

7. PLOS authors have the option to publish the peer review history of their article (what does this mean?). If published, this will include your full peer review and any attached files.

Reviewer #1: Yes: Thach Tran

Reviewer #2: No

---

## [Editor Report · Acceptance letter]

16 Jul 2020

PONE-D-19-30609R2 

Association between proton pump inhibitor use and risk of fracture: a population-based case-control study. 

Dear Dr. Sohn:

I'm pleased to inform you that your manuscript has been deemed suitable for publication in PLOS ONE. Congratulations! Your manuscript is now with our production department. 

Kind regards, 

on behalf of

Dr Robert Daniel Blank 

Academic Editor

PLOS ONE